# INSTRUCT2ACT: MAPPING MULTI-MODALITY INSTRUCTIONS TO ROBOTIC ARM ACTIONS WITH LARGE LANGUAGE MODEL

## ABSTRACT

Foundation models have significantly advanced in various applications, including text-to-image generation, open-vocabulary segmentation, and natural language processing. This paper presents `Instruct2Act`, a framework that leverages Large Language Models (LLMs) to convert multi-modal instructions to sequential actions for robotic manipulation tasks. Specifically, `Instruct2Act` uses LLMs to generate Python programs that form a comprehensive perception, planning, and action loop for robotic tasks. It uses pre-defined APIs to access multiple foundation models, with the Segment Anything Model (SAM) identifying potential objects and CLIP semantically classifying them. This approach combines the strengths of foundation models and robotic actions to transform complex high-level instructions into precise policy codes. Our approach is adaptable and versatile, capable of handling various instruction modalities and input types, and meeting specific task requirements. We validated the practicality and efficiency of our approach on robotic tasks including different tabletop and 6 Degree of Freedom(DoF) manipulation scenarios in both simulation and real-world environments. Furthermore, our zero-shot method surpasses many state-of-the-art learning-based policies in several tasks. The code for our proposed approach is available at https://anonymous.4open.science/r/Instruct2Act, providing a solid benchmark for high-level robotic instruction tasks with diverse modality inputs.

## 1 INTRODUCTION

Large Language Models (LLMs) such as GPT-3 (Brown et al., 2020), LLaMA (Touvron et al., 2023), and ChatGPT have made unprecedented progress in generating human-like text and understanding natural language instructions. These models exhibit impressive zero-shot generalization abilities, having been trained on extensive corpora and refined through human feedback. Subsequent work, such as Visual ChatGPT (Wu et al., 2023a), incorporates a various visual foundation models to enable visual drawing and editing through a prompt manager. Similarly, VISPROG (Gupta & Kembhavi, 2022) introduces a neuro-symbolic approach for intricate visual tasks, including image understanding, manipulation, and knowledge retrieval. Inspired by the immense potential of merging LLMs with multi-modality foundation models, we aim to develop comprehensive robotic manipulation systems. The question we pose is: Can we construct robotic systems akin to ChatGPT that support robotic manipulation, visual goal achievement, and visual reasoning?

Developing a versatile robotic system that can perform complex tasks in dynamic environments is a significant challenge in robotics research. Such a system needs to perceive its environments, choose appropriate robotic skills, and sequence them accordingly to accomplish long-term objectives. This requires the integration of various technologies, including perception, planning, and control, to allow the robot to function autonomously in unstructured environments. Inspired by the LLMs' impressive ability to generate simple Python programs from docstrings, CaP (Liang et al., 2022) directly generates the robot-centric policy code based on several in-context example language commands. However, its capabilities are limited to what the perception APIs can provide, and it struggles with interpreting longer and more complex commands due to the high precision requirements of the code.

To address these challenges, we propose a novel approach that utilizes multi-modality foundation models and LLMs to simultaneously execute perceptual recognition, task planning, and low-level

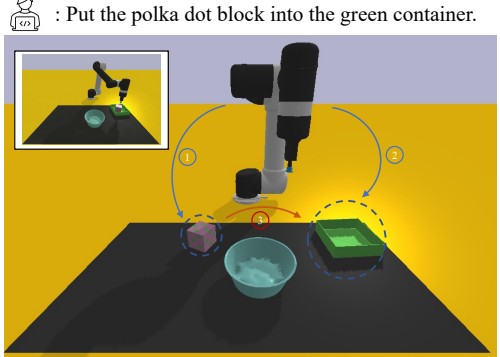 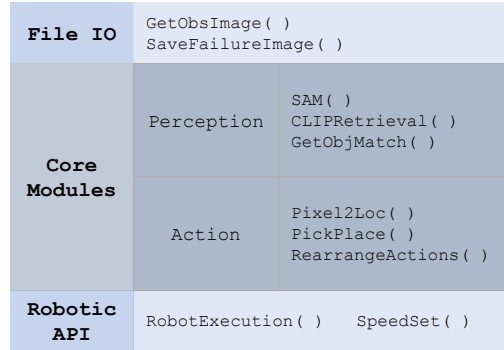

: Put the polka dot block into the green container.

(a) Robots are able to execute instructions that are provided as input in natural language.

(b) Module examples utilized in `Instruct2Act`. The modules' definitions are hierarchical and aligned with the robotic system design.

Figure 1: A robotic task (a) is executed through the invocation of several modules (b) in `Instruct2Act`.

control modules. Unlike existing methods such as CaP (Liang et al., 2022), which directly generates policy codes, our approach generates decision-making actions that can help reduce the error rate when performing complex tasks. Specifically, we use various foundation models, including SAM and CLIP, to accurately identify and classify objects in the environment. We then integrate this information with robotic skills to generate decision-making actions using LLMs.

We evaluate our proposed method across various domains and scenarios, encompassing simple object manipulation, visual goal achievement, and visual reasoning. Our framework offers a user-friendly, general-purpose robotic system and demonstrates robust performance on six representative meta-tasks from VIMABench (Jiang et al., 2022). Our proposed approach could serve as a solid baseline in the field of robotics research, contributing to the advancement of more intelligent and capable robots.

The contributions of our papers can be summarized as follows:

- **General-purpose robotic system.** We present a versatile robotic system, named `Instruct2Act`, which utilizes the in-context learning capabilities of LLMs and multi-modal instructions to generate mid-level decision-making actions from both natural language and visual instructions.

- **Flexible modality inputs.** This paper explores the use of unified modality instruction inputs in robotic tasks, including manipulation and reasoning, and introduces a flexible retrieval architecture capable of managing different types of instructions.

- **Strong zero-shot performance with minimal code overhead.** The proposed `Instrcut2Act` has demonstrated superior performance compared to current state-of-the-art learning-based policies, even without the need for fine-tuning. Moreover, the impact of using foundational models is relatively minor compared to methods that involve training from scratch.

## 2 RELATED WORKS

### 2.1 LANGUAGE-DRIVEN ROBOTICS

Language in robotics offers not only a user-friendly interface but also the potential for cross-task skill generalization and long-horizon task reasoning. As a result, instruction-based policies have been a popular area of research in robotics (Brohan et al., 2023; Shao et al., 2021; Shridhar & Hsu, 2018). Recently, with the emergence of multi-modality models, CLIPORT (Shridhar et al., 2022) has given Transporter (Zeng et al., 2021) the ability to understand semantics and manipulate objects by encoding text input through CLIP (Radford et al., 2021). Perceiver (Shridhar et al., 2023) extended the CLIPORT to the 3D domain by employing voxelized observation and action spaces in their Perceiver-Actor model. SayCan (Brohan et al., 2023) employed the 540B PaLM (Chowdhery et al., 2022) to accomplish zero-shot concept grounding. Language-planner (Huang et al., 2022a) utilized

two LLMs in their approach, where one was used for zero-shot planning generation and the other one was used for admissible action mapping. Inner monologue- (Huang et al., 2022b) enhanced their method by integrating closed-loop feedback, such as scene descriptors and success detectors, for performing robotic tasks. VIMA (Jiang et al., 2022) developed a large-scale benchmarking dataset by designing a multimodal prompts-conditioned framework. CaP (Liang et al., 2022) directly generates policy codes with detailed comments and context-specific examples to guide LLM output. LILAC (Cui et al., 2023) conducted a further investigation of the shared autonomy regime. Their approach involves a fusion of the correction signal from human instruction and the static controller with the original policy during inference. PaLM-E(Driess et al., 2023) built a large vision-language model for embodied agents by integrating the power of 540B PaLM (Chowdhery et al., 2022) and 22B ViT (Dehghani et al., 2023). Text2Motion (Lin et al., 2023) predicted the goal state and selected feasible actions using LLMs while considering geometric constraints. Socratic Models (Zeng et al., 2022) generated prompts in their approach by incorporating perceptual information into LLMs using vision-language models. TidyBot (Wu et al., 2023b) used LLMs to summarize the human's preferences given a few examples. Recently, VoxPoser (Huang et al., 2023) constructed a 3D value map to reflect the workspace and then used LLM interacting with a visual language model to update it, based on which robotic trajectories are generated with an additional planner. Our `Instruct2Act` framework achieves great flexibility while retaining expert domain knowledge by combining robotic primitive skills represented in API-style with the reasoning ability of LLMs.

## 2.2 FOUNDATION MODELS ON COMPUTER VISION TASKS

Several studies (Chen et al., 2020; Zhang et al., 2021) have utilized frozen pre-trained image encoders to improve the visual features extracted from images. And InternImage (Wang et al., 2022) adopted deformable convolutions in their large-scale visual foundation models for better image analysis. In addition, leveraging self-training and a massive dataset of 27M image-text pairs, GLIP (Li et al., 2022a)achieved strong zero-shot transfer ability. Segment Anything Model (SAM) (Kirillov et al., 2023), a segmentation foundation model trained on more than one billion mask samples, allows for zero-shot transfer to diverse tasks through prompt engineering. MaskCLIP+ (Liang et al., 2023) (Zhou et al., 2022) also discussed the zero-shot grounding ability of the visual foundation models. Furthermore, pre-trained LLMs have shown significant progress in text understanding and generation (Vaswani et al., 2017; Brown et al., 2020; Gao et al., 2023; Touvron et al., 2023). Such breakthrough in LLMs also benefits the VL tasks (Fu et al., 2021; Zhang et al., 2021). Recently, there have been explorations to combine the reasoning capacity of LLMs with the visual understanding ability of visual foundation models. VISPROG (Gupta & Kembhavi, 2022) utilizes in-context learning in GPT-3 to generate a program for new instruction and demonstrates the system's compositional visual reasoning ability. ViperGPT (Surís et al., 2023) leverages code-generation models to produce the results of language queries by means of composing foundation models into subroutines. Visual ChatGPT (Wu et al., 2023a) incorporates multiple visual foundation models and allows users to interact with ChatGPT through the proposed prompt manager, which allows multiple AI models reasoning ability with multi-steps. Similarly, the proposed `Instruct2Act` aims at endowing robotics the perception ability by incorporating advanced foundation models through the reasoning ability of LLMs.

## 2.3 FOUNDATION MODELS IN ROBOTICS

In addition to language-conditioned robotic manipulation, the use of foundation models has also led to significant advancements in robotics. LID (Li et al., 2022b) proposes a general approach to sequential decision-making that uses a pre-trained language model (LM) to initialize a policy network where goals and observations are embedded. R3M (Nair et al., 2022) explores how visual representations obtained by training on diverse human video data (Grauman et al., 2022) can enable data-efficient learning of downstream robotic manipulation tasks. Meanwhile, CACTI (Mandi et al., 2022) suggests a scalable framework for visual imitation learning that utilizes pre-trained models to map pixel values to low-dimensional latent embeddings for improved generalization ability. DALL-E-Bot (Kapelyukh et al., 2022) uses Stable Diffusion (Rombach et al., 2022) to generate goal scene images that function as guides for robot actions, providing a distinct approach compared to the aforementioned works. Recently, a few works (Wang et al., 2023; Kalithasan et al., 2022) proposed neuro-symbolic representations for robotic manipulation. However, they are limited to the close-set instructions and are not training-free. In contrast, our `Instruct2Act` framework employs visual foundation models as modular tools that can be invoked with APIs without the need for fine-tuning, thereby eliminating the need for data collection and training costs.

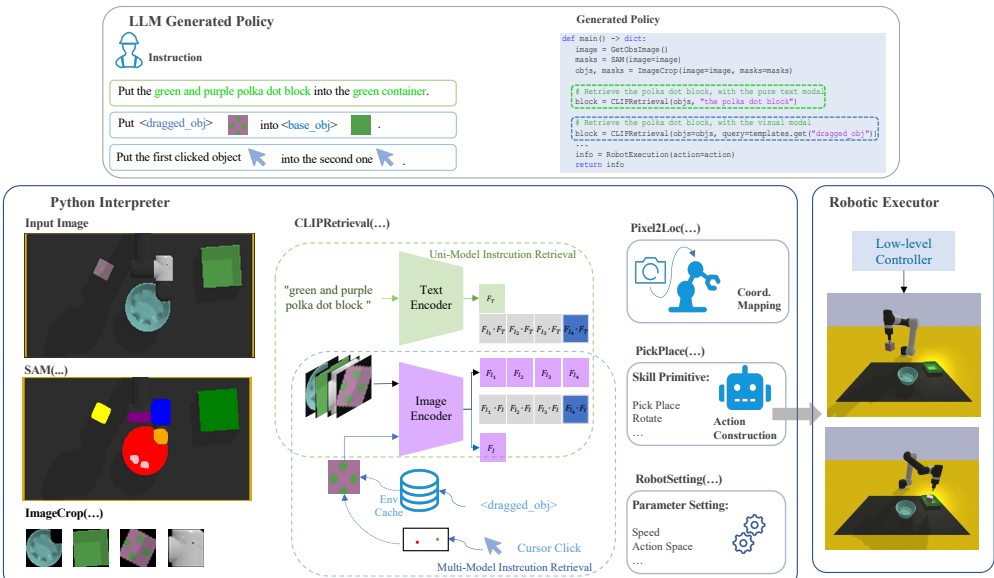

Figure 2: The paradigm of our proposed `Instruct2Act` framework. The framework employs an LLM to generate executable code that calls upon visual foundation models using APIs to recognize the environment. Once the object's semantic information is recognized, we generate plausible actions that are then sent to the low-level controller to execute the task. The instructions highlighted in green and blue represent pure-language and multimodal instructions, respectively.

## 3 METHODS

In general, `Instruct2Act` enables a robot arm to perform a series of actions based on user instructions and an image captured by a top-view camera. It is a language-based robotic system that generates perception-to-action codes using a large language model (LLM). The goal is to modify the state of objects in the environment to match the configuration described in the instructions. First, we explain how to control a robotic arm by LLM. Task-related variables, such as image crops from the task instruction and image-to-robot coordinate transformations, are stored in an environment cache **C** that can be accessed through an API. Next, we present the prompt design of `Instruct2Act`, including API definitions and in-context examples. Finally, we introduce the perception modules in detail.

### 3.1 HOW TO DRIVE ROBOTIC ARM BY LLM

To facilitate LLMs in completing robotic manipulation tasks, a designed prompt that guides the LLMs' generation is provided together with specific task instructions. The prompt includes application programming interfaces (APIs) and in-context examples to demonstrate their usage, which are introduced in the following section. The LLM's final input is a series of code APIs, usage examples, and task instructions. The LLM's output is a Python function in string format that can be executed by the Python interpreter to drive the robot's action.

Certain sections of the generated code lines invoke visual foundation models to extract pertinent visual information from the environment. This visual information, including the location and semantics of the target object, is then formulated as parameters used in the policy codes. After that, a mapping between the image space and the action space is established with a pre-defined transformation matrix. We also apply boundary clamping to avoid unintended actions.

Using LLM as the robotic driver offers several benefits. Firstly, the robotic policies generated by the LLM's API are highly adaptable and versatile, as in-context examples can be modified to guide LLM's behavior and adjust to new tasks. Secondly, by directly using visual foundation models, there's no need for data collection or training processes, and any enhancements in foundation models can improve action accuracy without additional costs. Lastly, the straightforward API naming and readable Python code make the generated policy code highly interpretable.

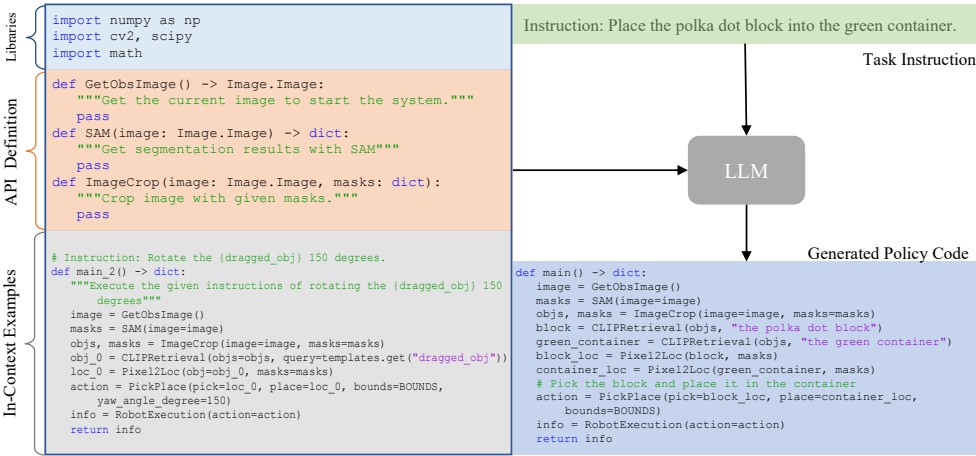

Figure 3: Robotic program generation process. The complete prompt consists of third-party import libraries, API definitions, and in-context examples.

## 3.2 PROMPTS FOR INSTRUCT2ACT

Fig 3 illustrates that a complete prompt should include essential information about third-party libraries, API definitions, and in-context examples. Importing third-party libraries allows the LLM to understand how APIs use parameter types defined by these libraries for calculations and even creating new functions. We showcase its effectiveness in Section 4.4. We also provide API definitions and descriptions, along with a few in-context examples to demonstrate their usage, similar to VISPROG (Gupta & Kembhavi, 2022). However, in our approach, we distribute and organize APIs based on robotic system information, as shown in Fig 1(b). Specifically, these APIs are classified according to their functionality within the robot system hierarchy, and this categorized information is provided in the prompt. For example, the `SAM()` function belongs to `Perception` module which is the second level core module in the robotic system. So we add `# Second Level:Core Modules` and `## Perception Modules` before introducing the `SAM()` API in the prompt. Moreover, unlike ViperGPT (Surís et al., 2023), which only offers function-level usage examples, we provide full-logical code examples invoking different modules similar to the approach of VISPROG (Gupta & Kembhavi, 2022). This choice is based on the notion that robotic tasks tend to be complex yet organized. In contrast to VISPROG (Gupta & Kembhavi, 2022), we design a prompt that provides coverage for all tasks and has fewer in-context examples. To encourage chain-of-thought reasoning, as done in (Kojima et al., 2022), we add the prompt `Think step by step to carry out the instruction` before the inserted specific task instruction. To avoid the LLM from generating too many redundant lines, we explicitly instruct it to only implement the `main()` function. Examples of complete prompts are available in Appendix A.8.

## 3.3 PERCEPTION WITH OFF-THE-SHELF FOUNDATION MODELS

`Instruct2Act` utilizes off-the-shelf visual foundation models, specifically, the Segment Anything Model (SAM) and CLIP models. These are accessed by the LLM via two designated APIs: `SAM()` and `CLIPRetrieval()`. These Python functions load and call the models to conduct the visual analysis. SAM outputs masks for all potential objects in the input image, based on these masks, object crops $I_i$ are extracted correspondingly. These crops are then encoded into $F_{I_i}$ using the CLIP image encoder and later utilized for the classification task. However, pre-trained visual models used directly on downstream tasks without any fine-tuning often suffer from incompleteness or incorrectness. To address these issues caused by the zero-shot paradigm, we insert processing modules between the output of the large model and the downstream tasks. To mitigate the effect of shadows, we apply a gray threshold filter followed by a morphological closing operation to fill up small holes before sending the image to the SAM. After SAM's segmentation operation, we perform a morphological opening operation to eliminate overly small holes or unconnected gaps. We also filter out masks with unreasonable sizes and reduce redundant mask output using Non-Maximum Suppression (NMS). For detailed discussions and visualizations of the processing steps, please refer to Appendix A.3.

### 3.4 FLEXIBLE INSTRUCTION MODALITY MANAGER

`Instruct2Act` is flexible and can handle inputs of multiple modalities, such as pure language and language-visual instructions, as depicted in Fig 2. We design a unified retrieval system that utilizes different types of queries to ensure the use of a unified architecture for both types of inputs.

**Pure-Language Instruction.** For pure language inputs, descriptive sentences are utilized to specify the target object and action. For example, a sentence such as `Put the green and purple polka dot block into the green container` can be employed. `Instruct2Act` utilizes the LLM to deduce that the robot needs to fetch the polka dot block from the environment. Therefore, the phrase `the green and purple polka dot block` acts as the query and is inputted into the CLIP text encoder to obtain the feature vector $F_T$. Finally, the similarity between the query embedding $F_T$ and the image crop features $F_{Ii}$ can localize the intended object precisely.

**Language-Visual Instruction.** For the multimodal inputs, the instruction uses an image to describe the target object or the target state. An example instruction is `Put <dragged_obj> into <base_obj>`. The placeholders in curly braces represent the corresponding images of individual objects, aligning with the LLM model's input format. Given the instruction, the LLM determines the placeholder strings to complete the query which is used to fetch the corresponding object image crop $I$ from the cache $\mathbf{C}$. Then the image crop $I$ is sent to the CLIP image encoder to obtain the feature vector $F_I$. We use $F_I$ to calculate the similarity with observation image feature vectors $F_{I_i}$.

Certain tasks require scene-level understanding, as demonstrated by the task of `Rearrange to this <scene>`. To fulfill this instruction, we first obtain every possible object and its corresponding feature vectors in the target scene image. We then use the Hungarian algorithm to determine the correspondences between the target scene image and the currently observed images.

**Pointing-Language Enhanced Instruction.** Pointing-language instructions are an effective alternative when the target object cannot be described using pure language instructions and providing image crops is impractical. Specifically, we adopt the cursor movement method from (Liu et al., 2023) and use cursor clicks to generate point prompts that guide the SAM's segmentation. Additional details can be found in Appendix A.4.

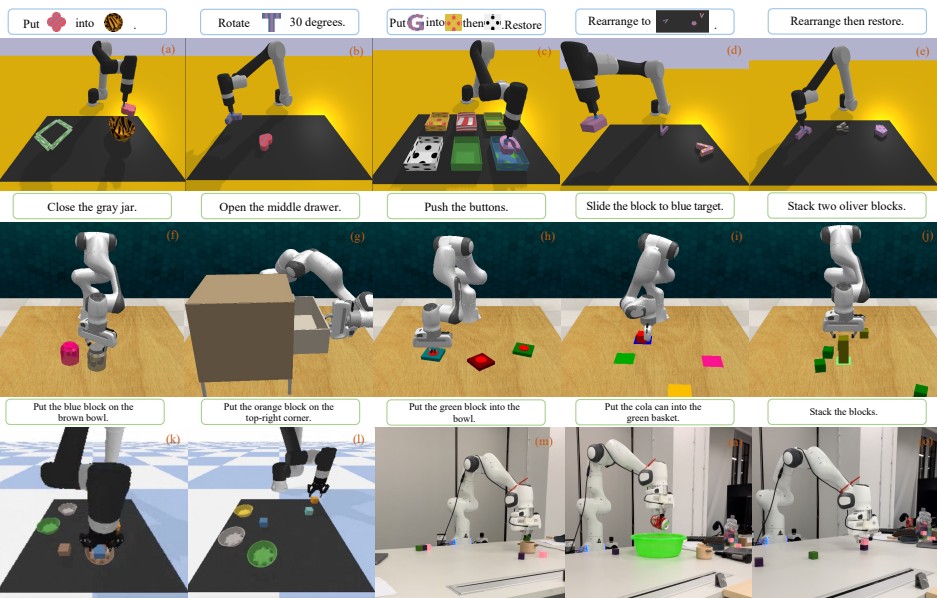

Figure 4: **Instruction-Conditioned Manipulation Tasks:** We conduct experiments on 17 tabletop manipulation tasks in VIMABench (Jiang et al., 2022)(a-e; only 5 shown), 5 six DoF manipulation tasks in RLBench (James et al., 2020)(f-i), and 5 tabtop manipulation tasks as done in CaP (Liang et al., 2022)(k-l; only 2 shown). We also demonstrate our approach with a Franka Panda on 5 real-world tasks(m-n; only 3 shown). The instructions in green and blue boxes are uni-modal and multi-modal instructions for the same task respectively.

## 4 EXPERIMENTS

### 4.1 EVALUATION TASK

**Multimodal Prompts Benchmark of Robot Manipulation.** We conducted multi-modality prompt manipulation tasks on VIMABench (Jiang et al., 2022), spanning from simple object manipulation to visual reasoning in the tabletop manipulation domain. These tasks are illustrated in Fig 4. VIMABench incorporates a 4-level generalization ability evaluation protocol, consisting of L1 placement, L2 combinatorial, L3 novel object, and L4 novel task generalization. Each level progressively deviates more from the training dataset distribution. For more detailed information about the evaluation setting, please refer to (Jiang et al., 2022).

**Textual Prompts Benchmark of Robot Manipulation.** VIMABench is currently the only available benchmark for multi-modality prompt manipulation tasks. To demonstrate the robustness and generalization ability of our versatile framework, we conducted experiments on additional textual prompt benchmarks. These benchmarks included 6-DoF manipulation tasks in RLBench (James et al., 2020) and 5 tabletop manipulation tasks similar to those in CaP (Liang et al., 2022), as shown in Fig 4. To make a fair comparison, we implemented an `Instruct2Act-Oracle` version, where we assume the object detection module returns the ground truth information. Additionally, we evaluated the performance of our approach on 5 real-world tasks using a Franka Panda robot.

### 4.2 IMPLEMENTATION DETAILS

In our experiments, we utilized two different types of Language Models (LLMs) to validate that the effectiveness of `Instruct2Act` is not significantly dependent on the choice of LLM. The first type is the *text-davinci-003* language model, accessed through the OpenAI API. This model is a fine-tuned variant of the InstructGPT (Ouyang et al., 2022) language model, optimized using human feedback. The second type is the LLaMA-Adapter (Gao et al., 2023; Zhang et al., 2023b), which we accessed through the user interface. This adapter is a lightweight adaptation of the original LLaMA models (Touvron et al., 2023). Our approach involved providing limited prompts to influence the output behavior of the language models, without any training or fine-tuning. Although the language models may occasionally generate incomplete or incorrect code, such as missing brackets, punctuation, or mismatched cases, the Python Interpreter can detect these errors and prompt us to generate new code.

For open-vocabulary segmentation, we used open-sourced models such as SAM ViT-H and CLIP ViT-H-14 for classification. To evaluate task success rates, we conducted experiments on 150 instances for each representative meta-task. For each meta-task, we selected three random seeds to calculate average success rates. The VIMABench simulator determined task success by comparing the final states with the configurations specified in the instructions. These experiments were conducted on an NVIDIA 3090Ti GPU. Additionally, we directly utilized the experiment result of different baselines (20M parameter version) in VIMABench from (Jiang et al., 2022).

To extend the original tabletop manipulation skills and evaluate the agent with 6 DoF tasks, we used the RGB-D camera as the input sensor following PerAct (Shridhar et al., 2023) and made the following medications. For simple 3D tasks, such as stacking objects, we introduced an additional parameter to indicate the current occupancy. For more complex tasks, such as opening a drawer, we added some heuristic movements to ease the execution. Further details regarding the 6 DoF manipulations and real-world experiments can be found in Appendix A.5.

### 4.3 RESULTS

**Results on Multimodal Prompts Benchmark.** We demonstrate the effectiveness of our proposed method through experiments on various generalization tasks, including L1, L2, L3, and L4 levels of VIMABench. Fig. 5 displays the results of these methods. Compared with Gato (Reed et al., 2022), Flamingo (Alayrac et al., 2022), and Decision Transformer (DT) (Chen et al., 2021), our method consistently yielded positive results

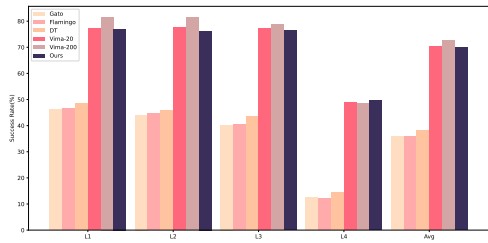

Figure 5: Evaluation results on VIMABench.

across all levels and tasks. In the case of VIMA, our framework achieves comparable performance without the necessity of training, further showcasing its versatility and effectiveness. In contrast, VIMA requires about 650K training trajectories and multiple GPUs for its training process.

**Results on Other Textual Prompts Benchmarks.** Besides the multimodal prompts benchmark, we evaluate our proposed approach on several textual prompts benchmarks to confirm its generalization ability. Table 1 shows that we achieve a 3.8% SR gain compared to the PerAct (Shridhar et al., 2023) method. For the CaP benchmark, we obtain a 1.2% SR gain using significantly fewer tokens compared to CaP (Liang et al., 2022).

| Benchmark | Method | SR(%) | Benchmark | Method | SR(%) | Token |
|---|---|---|---|---|---|---|
| RLBench | PerAct | 59.2 | CaP | CaP | 84.8 | 3919 |
| | Ours | 63 | | Ours-Oracle | 86 | 1600 |

Table 1: Evaluation results on textual prompts benchmarks.

## 4.4 ABLATION STUDIES

**Effectiveness of Prompt Components.** We conducted an investigation into the effectiveness of each prompt component in generating policy code. To thoroughly validate this effectiveness, we created two additional task sets: the Reformatted Instruction set (where instructions such as representing angles in degrees were replaced with radians) and the Unseen Tasks set (where information about RR was removed from the prompts). Table 2 shows that when all prompt components are provided to the LLM, the agent achieves the best performance. Due to the absence of full-logic code examples, as in ViperGPT, the agent experiences a significant decline in performance, *e.g.* SR drop from 83% to 28%. We attribute this to the complexity of robotic tasks, which often involve multiple steps and variables during execution. The lack of demonstrations using complete policy codes poses challenges for the LLM in generating accurate codes. Additionally, we discovered that information on Libraries and APIs proves advantageous when dealing with both novel task instructions and tasks that have not been previously encountered.

| Library | API | Full Examples | Normal Tasks | Reformated Instruction | Unseen Tasks | Average |
|---|---|---|---|---|---|---|
| ✓ | ✓ | | 33 | 30 | 20 | 28 |
| ✓ | | ✓ | 80 | 75 | 60 | 72 |
| | ✓ | ✓ | 85 | 75 | 80 | 80 |
| ✓ | ✓ | ✓ | 85 | 80 | 85 | 83 |

Table 2: Ablation on the effectiveness of prompt components.

**Different Foundational Models.** We conducted experiments using different visual foundation models to assess their impact on performance. Specifically, we replaced the original models with SAM-Base and SAM-Large for the semantic segmentation module, and Base-16 and Large-14 for the CLIP model. Considering the deployment in the robotic domain, we further provided the results with FastSAM (Zhao et al., 2023) and MobileSAM (Zhang et al., 2023a). The results, displayed in Table 3, consistently indicate that larger foundation models improve the performance of Instruct2Act. This finding suggests that our approach could benefit from even more powerful foundation models in the future. Moreover, since the visual models are accessed via APIs, they can be easily substituted with other visual foundation models. We provide a task-level result in Appendix A.6.

| CLIP | H-14 | H-14 | H-14 | H-14 | H-14 | B-16 | L-14 |
|---|---|---|---|---|---|---|---|
| **SAM** | Base | Large | Huge | FastSAM | MobileSAM | Huge | Huge |
| **SR(%)** | 75.3 | 83 | 84.4 | 83.2 | 76.2 | 70.4 | 76.5 |

Table 3: Ablation on the visual foundation models.

**Effectiveness of Processing Modules.** Table 4 presents ablation studies that validate the effectiveness of the proposed processing modules, including image pre-processing and mask post-processing. The absence of any processing methods leads to significant performance degradation, with only $51\%$ SR, in line with the findings of the analysis in Section 3.3. Mask post-processing directly boosts the success rate to 83.0%, while further incorporation of image pre-processing achieves the peak success rate of $84.1\%$. It is worth noting that using only image pre-processing leads to no improvement in performance and may even cause degradation in some tasks. This could be attributed to the fact that the current heuristic method would inevitably destroy some features on target objects since

some tested objects also have parts with a color similar to the background. In other words, image pre-processing offers potentially better segmentation results at the cost of losing some visual parts.

| Process Method | | VM | SU | RT | RA | RR | POR | Average |
|---|---|---|---|---|---|---|---|---|
| Image | Mask | | | | | | | |
| | | 70.4 | 34.6 | 88.6 | 41.7 | 15.9 | 54.7 | 51.0 |
| ✓ | | 69.7 | 33.9 | 87.7 | 40.9 | 14.9 | 52.9 | 50.0 |
| | ✓ | **91.7** | 78.2 | 97.4 | 72.9 | **69.5** | 88.3 | 83.0 |
| ✓ | ✓ | 91.6 | **80.8** | **97.8** | **78.4** | 69.1 | **87.2** | **84.1** |

Table 4: Ablation studies on the effectiveness of the proposed processing modules. The success rates were calculated as the average of performance across three generalized levels. The tasks included Visual Manipulation (VM), Scene Understanding (SU), Rotation (RT), Rearrange (RA), Rotate and Restore (RR), and Pick in Order then Restore (POR).

**Comparison between Different LLMs.** We demonstrate that our proposed `Instruct2Act` is potentially effective with open-source LLM, in addition to the commercial ChatGPT. Specifically, we use the

| VM | Direct | More Trail | Filtering |
|---|---|---|---|
| SR(%) | 72.5 | 77.5 | 85.5 |

Table 5: Results with LLaMA-Adapter.

LLaMA-Adapter (Gao et al., 2023) and choose Visual Manipulation(VM), reporting the success rate on 40 instances in Table 5. Remarkably, our `Instruct2Act` already achieves plausible performance with the LLaMA-Adapter's original output. Furthermore, our performance improves to 77.5% when we increase the number of generation trials when the Python interpreter raises an exception. And comparable results can be achieved by using basic filtering, such as re-generating when the environment cache usage is missed in the code.

**Flexibility and Robustness of `Instruct2Act`.** We demonstrate the flexibility and robustness of `Instruct2Act` on the Out-of-distribution (OOD) instructions. Specifically, we evaluate our approach in three scenarios: Human Intervention(HI), Missing Characteristics(MC), Synonym Replacement(SR), and Instruction Reformat(IR). We conducted the experiments on the Task Rotation and reported a success rate of 20 runs. As shown in Table 6, VIMA failed to handle HI and SR. In contrast, `Instruct2Act` can ground the instruction correctly. Furthermore, it can use the imported 3-rd library such as `numpy` to convert the radians to degrees before passing to the action function.

| OOD Instruction | Examples | Ours | VIMA |
|---|---|---|---|
| Human Intervention | "Cancel the task, stop!" | 100% | 0% |
| Missing Characteristic | Drop one characteristic from "Rotate" randomly. | 100% | 100% |
| Synomym Replacement | Replace "Rotate" with "Spin". | 100% | 95% |
| Reform Task | Represent the angle degrees with radians. | 95% | 0% |

Table 6: Flexibility of `Instruct2Act` with OOD instructions.

**Further Analysis.** We further conducted the error analysis across different tasks and defined five error types accordingly. The detailed error analysis can be found in Appendix A.1. Besides, we provide a further discussion on the prompt designs within the LLM-based API-tool domain, which can be found in Appendix A.2. Deeper analyses with full evaluation results tables are in Appendix A.7.

**Limitations.** Our method is presently limited by the basic action primitives, such as Pick and Place. However, we intend to expand our approach by incorporating more complex actions through API extensions in future work. Additionally, we have only tested our method with simple real-world manipulation tasks, and will investigate more complex real-world applications in the future.

## 5 CONCLUSION

We proposed a `Instruct2Act` framework to utilize LLM to map multi-modality instructions to sequential actions in the robotics domain. With the LLM-generated policy codes, various visual foundation models are invoked with APIs to gain a visual understanding of the task sets. To mitigate the gaps in the zero-shot setting, some processing modules are plugged in. Extensive experiments in the simulation and real-world setting verify that `Instruct2Act` is effective and flexible in robotic manipulation tasks.

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

## A APPENDIX

### A.1 TASK ERROR ANALYSIS

We conducted an error analysis on the generated policy codes. The code-style policy in `Instruct2Act` provides a robust mechanism for pinpointing error/failure sources. We manually go over 100 examples on the first 10 tasks of VIMABench, where each task has 10 different policy codes. Using criteria inspired by VISPROG (Gupta & Kembhavi, 2022), we categorized errors as API misuse; Attributes hallucination (using the attributes that do not exist); Syntax error(including referring the inaccessible parameters); Inaccurate Perception(detection and segmentation); Limited API's ability (correct logic codes while wrong execution results except the perception reason). Among these factors, API misuse, attributes hallucination and syntax error can be taken as policy generation errors while the other two are inherent in the API implementation. As shown in Table 7, the policy generation error rate in `Instruct2Act` is much lower than in ViperGPT, indicating the effectiveness of in-context full policy code examples which help the LLM understand the usage of different APIs and construct a complete policy code.

| Type of Error | Description of Error | Ours | ViperGPT |
|---|---|---|---|
| API misuse | Use the incorrect API during execution | 4% | 16% |
| Attributes hallucination | Use the nonexistent attribute | 2% | 4% |
| Syntax error | Program syntax error rasing from interpreter | 3% | 14% |
| Inaccurate Perception | Inaccurate detection or segmentation result | 6% | 5% |
| Limited API's ability | Inaccuracy during the API's execution | 5% | 6% |
| Correct | Execute successfully | 76% | 55% |

Table 7: Error distribution across 100 task runs on VIMABench.

Since the tasks on VIMABench are diverse and focus on different robotic abilities, we conduct further error analysis on the task-specific level. We chose three representative tasks from the VIMABench: Visual Manipulation (VM, one of the simplest tasks), Rearrange then Restore (RR, requiring multi-steps), and Twist (TW, requiring stronger reasoning ability on instructions).

As shown in Table 8, the task Visual Manipulation has the least errors in code generation since it requires few task understanding and almost no new function generation ability. For Task Rearrange then Restore, the predominant source of error is 'Inaccurate Perceotion'. This can largely be attributed to the scene in that the task often encompasses multiple objects, some of which possess analogous textures, posing challenges for CLIP's differentiation capabilities. Meanwhile, for Task Twist, 'Limited API's ability' emerges as the primary error cause. This task necessitates the policy to first deduce the intended rotation angle from template images prior to task execution. With current heuristic methods, obtaining a precise and reliable value proves challenging.

| Type of Error | VM | RR | TW |
|---|---|---|---|
| API misuse | 2% | 8% | 6% |
| Attributes hallucination | 2% | 4% | 4% |
| Syntax error | 0% | 4% | 4% |
| Inaccurate Perception | 6% | 12% | 8% |
| Limited API's ability | 0% | 2% | 24% |
| Correct | 90% | 70% | 54% |

Table 8: Error distribution across 3 meta tasks on VIMABench.

### A.2 FURTHER DISCUSSION ON PROMPT DESIGN

Based on the evaluation results and error analysis, we further elucidate our findings regarding the prompt design in API usage with LLM. Current methodologies can be categorized into three primary groups:

**Pure Text Prompt with Online Low-level Code Generation:** This approach instructs the LLM to autonomously produce the requisite codes. Notably, while this method offers tremendous flexi-

bility, it consumes a significant amount of tokens. Additionally, the LLM's inherent limitations in spatial reasoning render it ineffective for certain tasks. The representative work in this category is CaP (Liang et al., 2022).

**API-Only Information:** Here, the LLM is granted a high degree of latitude in selecting the API tools. Nonetheless, it occasionally struggles with crafting multi-step tool utilization for intricate tasks. Furthermore, inherent randomness in the LLM can lead to variable naming inconsistencies. ViperGPT (Surís et al., 2023) exemplifies this category.

**Logic-Only Codes:** This approach solely offers logic codes, potentially restricting the LLM's capacity to architect new code logic. Consequently, this might curtail its adaptability to novel tasks. VISPROG (Gupta & Kembhavi, 2022) epitomizes this group.

In conclusion, the selection of prompt design should be tailored to the specific requirements of the task at hand. `Instruct2Act` amalgamates both API information and complete logical code samples, ensuring it caters to structured robotic task settings while preserving the flexibility to adapt to new tasks.

### A.3    PROCESSING MODULE IN INSTRUCT2ACT

When pre-trained models are utilized directly on downstream tasks without any fine-tuning, they inevitably suffer from problems like incompleteness or incorrectness. In light of this, it is advisable to insert processing modules or adapters between the output of the large model and the downstream tasks which can effectively tackle the issues caused by this zero-shot paradigm. The entire loop of execution for robot tasks comprises three modules, namely, perception, planning, and execution. We have curated various processing programs for each of these modules designed for tabletop manipulation domains.

**Image Pre-Processing** In the case of zero-shot SAM outputs, a significant challenge is distinguishing the target object from shadow regions that might appear within the image. As object shadows cannot be grasped, their presence poses additional difficulty for robot grasping tasks. Furthermore, tabletop manipulation domains typically involve camera placement above the robotic arm, leading to large shadows cast onto the operating table. To account for this, we adopted a simple yet efficient image preprocessing methodology: to mitigate the effect of shadows, we employed the gray threshold filter followed by the close morphological operation to fill in small gaps that the filter might produce.

**Mask Post-Processing** In the zero-shot setting, it is observed that the SAM produces multiple segments (e.g. masks) that may be discontinuous or discrete. There may also be detected objects with missing parts, or holes inside, as shown in Fig 6-C. These misleading semantic mask outputs will inevitably confuse the subsequent modules and greatly challenge the robot grasping task. In order to address the given problem, we have developed a set of processing modules to work with the SAM output. The modules include the following methods:

- We apply a filtering process on the output based on the mask's size. This process removes any output that is clearly not part of the target object. Such output may include objects that cannot be moved, like tables or patterns on the target object.

- A dilation operation is used to effectively eliminate unrealistic small holes or unconnected gaps. To avoid significant changes to the mask's size due to dilation, we then use an erosion operation. These two procedures combine to create what is known as the opening morphological operation.

- In some instances where there are multiple segmentation outputs for a single object, we employ the Non-Maximum Suppression (NMS) operator to reduce redundant mask output.

### A.4    FURTHER DISCUSSION ON INSTRUCTION MODALITIES ON VIMABENCH

Pure-Language Instruction In addition to the original multimodal prompt instruction in the VIMABench, we extract object descriptions from the simulator and create a task prompt utilizing natural language that is more commonly used by actual users in their day-to-day lives.

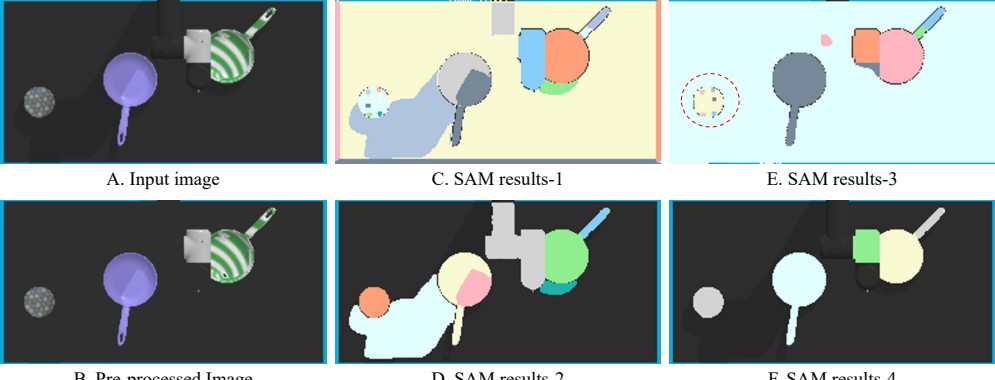

| A. Input image | C. SAM results-1 | E. SAM results-3 |
| B. Pre-processed Image | D. SAM results-2 | F. SAM results-4 |

Figure 6: The segmentation outputs w / wo the processing methods. Fig A is the original image which is produced by a top-down camera. Fig B is the image with the proposed image pre-processing pipeline. Fig- $\{C, D, E, F\}$ are the SAM results directly on the original image, only with the mask post-processing, only with image pre-processing, and with both processing modules respectively. The portion highlighted by the red circle demonstrates the redundant output of the SAM without additional processing applied to a texture object.

Pointing-Language Enhanced Instruction When utilizing the pointing-language mode of `Instruct2Act`, the system will display the initial task instruction and observation image. Next, the user will select the target objects by clicking on them with the cursor. These click points will then act as point prompts to guide the SAM's segmentation process. We evaluated this mode on Visual Manipulation (VM), Rotate(RT), and Pick in Order the Restore (POR), averaging the results over 100 instances with one seed.

| Instruction Modal | VM | RT | POR |
|---|---|---|---|
| Original (Visual-Language Instruction) | 91.3 | 98.2 | 85.2 |
| Pure-Language Instruction | 86.7 | 94.6 | 63.0 |
| Pointing-Language Enhanced Instruction | 90.7 | 98.0 | 97.5 |

Table 9: Results with pointing-language enhanced instruction.

As shown in Table 9, the performance of multimodal instructions is generally better than that of single-modal instructions. We believe that this is because the former provides a more comprehensive range of information to the model, thereby reducing difficulties for the robot when attempting to reason about the current execution scenario. Moreover, it achieves better results with the pointing-language enhanced mode than with the pure-language instruction mode. This could be due to the stronger prior information provided by the user's click operations.

## A.5 EVALUATION TASK DESCRIPTION

### A.5.1 TASKS ON VIMABENCH

VIMABench (Jiang et al., 2022) presents a 4-level evaluation protocol that progressively increases in difficulty for trained agents. Level 1 (L1) placement generalization randomly arranges the placement of target objects, whereas L2 combinatorial generalization generates new combinations of target materials and object descriptions. L3 novel object generalization tests our agents' ability to generalize to novel materials and objects. L4 novel task generalization needs the agent to ground and execute tasks that are previously unseen. We describe the tasks in the order of their corresponding task index which is identical to the VIMABench.

- Visual Manipulation. The agent is required to pick a specific object and place it into a specified container. The agent needs to first recognize the target objects aligning with the task instruction specified by the natural language description or by the image pattern.

- Scene Understanding. The agent must first identify the target object with the described texture by grounding the natural language description and the scene image simultaneously and then put the target object into the container with a specific color.

- Rotation. The agent is required to rotate a specific object by certain degrees along the z-axis.

- Rearrange. The agent is required to rearrange the target objects to reach the goal configuration. The agent needs to identify the possible existing distractors and move them away to avoid position conflicts.

- Rearrange then restore. The agent is required to restore the object placements after the rearrangement operations.

- Novel adjectives understanding. The agent needs to ground new adjectives by comparing the size or the textural of objects at first.

- Novel nouns. Similar to the above but with novel nouns.

- Novel nouns and adjectives. The combination of Novel adjectives understanding and novel nouns understanding.

- Novel concept understanding. The agent needs to infer what is the exact angle to rotate from the prompt.

- Follow the motion. The agent is required to place the target object to match the pose in the corresponding video frame.

- Stack the blocks. Similar to the above.

- Sweep without exceeding. The agent needs to sweep target objects inside the specified region without exceeding the constraint.

- Sweep without touching. Similar to the above except that the agent needs to not touch the red line.

- Same texture. The agent needs to pick objects with the same texture as the descriptor in the prompt and place them in the container.

- Same shape. All objects with the same shape as the container should be placed in the container.

- Manipulate the old neighbor. The agent should first place the target object into the container, then put one of its old neighbors into the same container.

- Pick in order then restore. The agent is required to pick and place the target object sequentially into different containers and finally restore it to the initial container.

### A.5.2 TASKS ON RLBENCH

RLBench James et al. (2020) is one of the most popular vision-guided manipulation benchmarks. Following the PerAct (Shridhar et al., 2023), we evaluated our methods on the subset of the RL-Bench. Similarly, we also use the RGB-D camera mounted at the robot's wrist as our input sensor.

- Close jar. The robot needs to put the lid on the jar with a specific color. We use the depth information to distinguish the lid and the jar.

- Push buttons. The agent is required to push the colored buttons in the specified sequence. The depth difference is used to estimate the button size and we heuristically set the button-pressing action.

- Slide blocks. The agent needs to slide the block to one of the colored square targets. And we used similar methods as the task *PushButtons*.

- Stack blocks. The agent needs to stack N blocks of the specified color on the green platform. We maintain an addition parameter to indicate the current occupancy, e.g. below 1.5 cm is occupied for Position A due to the existence of Block 1.

- Open drawer. The agent is required to open one of the three drawers: top, middle, and bottom. We first heuristically move the robotic arm and let the camera on the wrist face the drawer. Then we use the image from the wrist camera as the main input.

### A.5.3 Tasks on CaP Benchmarks

To compare with CaP (Liang et al., 2022), we also evaluate our method on the tasks provided in their open-source projects. To mark a fair comparison, we use the ground truth information of the objects' existence from the simulator as done in CaP.

- Pick and place. Pick up the object and place it on the receptacle bowl.
- Put in corner. Put all the blocks on the corner/side.
- Directional corner. Pick up the block to the defined direction of the specific receptacle bowl and place it on the defined corner/side.
- Stack blocks. Stack all the blocks.
- Put in line. Put all blocks in a specific line.

### A.5.4 Tasks on real-world setting

The real-world experiments use a Franka Panda manipulator with a parallel gripper. or perception, we use a RealSense camera mounted on the gripper. The extrinsic calibration between the camera and the robot base frame is computed through ARUCO ROS[1] in previous. We use the Python controller provided by Deoxys[2]. In real-world experiments, we conducted five tasks. And we provide real-world demos in the supplementary material.

- Rotate. Rotate a specific object with the given angle.
- Place the block. Place the block with the given color into the receptacle.
- Stack blocks.
- Stack blocks then restore. First stack the blocks, then restore the moved one to its original position.
- Place the can. Place a cola can into the receptacle. This is used for testing the open-vocabulary ability of our method.

### A.6 Task-level Evaluation Results with Different Visual Foundation Models

As we can see from Table 10, with a stronger off-the-shelf perception model, the overall performance could get better. However, we also noticed that with about 10% percent parameters, the FastSAM could have a comparable performance. This could partly be due to the fact that the tested environment is relatively structured and with fewer visual reasoning challenges. In such a case, FastSAM could be a better choice since the computation cost is much lower.

| Method | FastSAM | MobileSAM | SAM |
|---|---|---|---|
| Parameters | 68M | 10M | 641M |
| Visual manipulation | 89.2 | 83.2 | 91.3 |
| Scene understanding | 82.0 | 76.3 | 81.4 |
| Rotate | 97.9 | 95.3 | 98.2 |
| Rearrange | 76.7 | 70.6 | 78.5 |
| Rearrange then restore | 70.2 | 65.5 | 72.0 |
| Pick in order then restore | 83.2 | 66.4 | 85.2 |
| Average | 83.2 | 76.2 | 84.4 |

Table 10: Task-level ablation on the visual foundation models. The ViT-H is chosen as the default CLIP model.

---

[1] https://github.com/pal-robotics/aruco_ros
[2] https://github.com/UT-Austin-RPL/deoxys_control

A.7    FULL EVALUATION RESULT TABLES

This section contains more detailed tables corresponding to the results shown in Figure 5 and Table 1.

Based on the evaluation results, several conclusions can be drawn:

- Nero-symbolic representation demonstrates a clear advantage in long-horizon tasks. This is attributed to the utilization of a more compact representation of action sequences, where multiple-step actions can be represented by API calls. The superiority of this approach is validated through the evaluation of VIMABench Task 05 *RearrangethenRestore* and Task 17 *PickInOrderthenRestore*, where our method outperforms previous state-of-the-art models significantly.

- The utilization of Python code for implicit memory module and position reasoning simplifies task execution. Memory (such as tracking manipulated objects) and position reasoning (such as determining the relative position of objects), are typically challenging to model in the field of robot learning. However, we find that these difficulties can be overcome with the incorporation of Python code invoked by LLM. This is demonstrated by the successful completion of Task 16 *ManipulateOldNeighbours*. Moreover, the deterministic usage of Python code helps prevent the occurrence of hallucination phenomena in LLM, as evidenced by Task *directional corner* on CaP.

- The integration of LLM in Instruct2Act serves as an effective task reasoner. Through the use of our designed prompts, our method exhibits the ability to synthesize new task skills for novel tasks, as indicated by the L4 evaluation results on VIMABench 14. Additionally, by leveraging visual foundation models, the system is capable of extracting visual information efficiently. This, combined with the reasoning abilities of LLM, enables the agent to comprehend novel and complex tasks, as exemplified by Task 09 *Twist*.

| Method | 01 | 02 | 03 | 04 | 05 | 06 | 07 | 09 | 11 | 12 | 15 | 16 | 17 |
|---|---|---|---|---|---|---|---|---|---|---|---|---|---|
| Gato-20M | 61.5 | 62 | 32.5 | 49 | 38 | 46 | 60 | 5 | 68 | 83 | 47 | 46.5 | 2 |
| Flamingo-20M | 63 | 61.5 | 55 | 50 | 42.5 | 41.5 | 58 | 6 | 62 | 83 | 44 | 38.5 | 1 |
| DT-20M | 60.5 | 64 | 50.5 | 44 | 41 | 48 | 61.5 | 7 | 85 | 84 | 44.5 | 39 | 2.5 |
| VIMA-20M | 100 | 100 | 100 | 100 | 59.5 | 100 | 100 | 13.5 | 74 | 72.5 | 96.5 | 39.5 | 47.5 |
| VIMA-200M | **100** | **100** | 99.5 | **100** | 56.5 | **100** | **100** | 18 | **77** | **93** | **97** | **76.5** | 43 |
| Ours | 91.3 | 81.4 | 98.2 | 78.5 | **72** | 82 | 88 | **42** | 72 | 68 | 78 | 64 | **85.2** |

Table 11: VIMABench L1 level generalization results. The number in the method column indicates the controller parameter count. Integers in the first row refer to the order described in Appendix A.5.1.

| Method | 01 | 02 | 03 | 04 | 05 | 06 | 07 | 09 | 11 | 12 | 15 | 16 | 17 |
|---|---|---|---|---|---|---|---|---|---|---|---|---|---|
| Gato-20M | 44 | 51.5 | 39 | 51 | 38.5 | 47.5 | 52.5 | 6 | 65.5 | 84 | 52.5 | 40.5 | 1 |
| Flamingo-20M | 48.5 | 49 | 55.5 | 48 | 42.5 | 46.5 | 52 | 6 | 66 | 82 | 47.5 | 37 | 0.5 |
| DT-20M | 50.5 | 49.5 | 53 | 44.5 | 43.5 | 47 | 46 | 8 | 83.5 | 80 | 46.5 | 41 | 2.5 |
| VIMA-20M | 100 | 100 | 100 | 100 | 61 | 100 | 100 | 16.5 | 75.5 | 75 | 96 | 37.5 | 47.5 |
| VIMA-200M | **100** | **100** | 99.5 | **100** | 54.5 | **100** | **100** | 17.5 | **77** | **93** | **98.5** | **75** | 45 |
| Ours | 91.7 | 80.8 | 97.8 | 74.9 | **69.5** | 81 | 86 | **44** | 70.5 | 65 | 80 | 66 | **84** |

Table 12: VIMABench L2 level generalization results. The number in the method column indicates the controller parameter count. Integers in the first row refer to the order described in Appendix A.5.1.

| Method | 01 | 02 | 03 | 04 | 05 | 06 | 07 | 09 | 11 | 15 | 16 | 17 |
|---|---|---|---|---|---|---|---|---|---|---|---|---|
| Gato-20M | 46.5 | 55 | 44.5 | 57 | 31.5 | 47.5 | 51.5 | 2.5 | 72.5 | 30.5 | 44 | 0 |
| Flamingo-20M | 47 | 54.5 | 53 | 55 | 36 | 42.5 | 48 | 6.5 | 70 | 33 | 41.5 | 0 |
| DT-20M | 50 | 60.5 | 56.5 | 48 | 33.5 | 51 | 46 | 6.5 | 92.5 | 32.5 | 43.5 | 1.5 |
| VIMA-20M | 98 | 100 | 100 | **98.5** | 55.5 | 100 | **99.5** | 15 | 88.5 | **99.5** | 44 | 29.5 |
| VIMA-200M | **99** | 100 | 100 | 97 | 54.5 | **100** | 99 | 17.5 | **90.5** | 97.5 | 46 | 43.5 |
| Ours | 91.8 | 80.2 | 97.4 | 81.8 | **65.8** | 79 | 89 | **38** | 71 | 78 | **62** | **82** |

Table 13: VIMABench L3 level generalization results. The number in the method column indicates the controller parameter count. Integers in the first row refer to the order described in Appendix A.5.1.

| Method | 08 | 10 | 13 | 14 |
|---|---|---|---|---|
| Gato-20M | 20.5 | 0 | 0 | 29 |
| Flamingo-20M | 21 | 0 | 0 | 27.5 |
| DT-20M | 20.5 | 0.5 | 0 | 36 |
| VIMA-20M | 100 | 0 | 0 | 95.5 |
| VIMA-200M | **100** | 0 | **0** | 94.5 |
| Ours | 84 | **35** | 0 | 80 |

Table 14: VIMABench L4 level generalization results. The number in the method column indicates the controller parameter count. Integers in the first row refer to the order described in Appendix A.5.1.

| Model | close jar | open drawer | push buttons | slide blocks | stack blocks |
|---|---|---|---|---|---|
| PerAct | 60 | 80 | 48 | 72 | 36 |
| Ours | 55 | 60 | 70 | 75 | 55 |

Table 15: Evaluation results on RLBench.

| Model | pick and place | put in corner | directional corner | stack blocks | put in line |
|---|---|---|---|---|---|
| CaP | 88 | 92 | 72 | 82 | 90 |
| Ours-Oracle | 100 | 95 | 90 | 90 | 55 |

Table 16: Evaluation results on CaP.

## A.8 FULL PROMPTS IN INSTRUCT2ACT

Listing 1: An example of a full prompt in `Instruct2Act`

```
THIRD-PARTY TOOLS:
------
You have access to the following tools:

# Libraries
from PIL import Image
import numpy as np
import scipy
import torch
import cv2
import math
from typing import Union

IMPLEMENTED TOOLS:
------
You have access to the following tools:

# First Level: File IO
templates = {} # dictionary to store and cache the multi-modality
    instruction
# possible keys in templates: "scene", "dragged_obj", "base_obj"
# NOTE: the word in one instruction inside {} stands for the visual part
    of the instruction and will be obtained with get operation
# Example: {scene} -> templates.get('scene')
BOUNDS = {} # dictionary to store action space boundary

def GetObsImage(obs) -> Image.Image:
    """Get the current image to start the system.
    Examples:
        image = GetObsImage(obs)
    """
    pass

def SaveFailureImage() -> str:
    """Save images when execution fails
    Examples:
        info = SaveFailureImage()
    """
    pass

# Second Level: Core Modules
## Perception Modules
def SAM(image: Image.Image) -> dict:
    """Get segmentation results with SAM
    Examples:
        masks = SAM(image=image)
    """
    pass

def ImageCrop(image: Image.Image, masks: dict):
    """Crop image with given masks
    Examples:
        objs, masks = ImageCrop(image=image, masks=masks)
    """
    pass

def CLIPRetrieval(objs: list, query: str | Image.Image , pre_obj1: int =
     None, pre_obj2: int = None) -> np.ndarray:
    """Retrieve the desired object(s) with CLIP, the query could be
        string or an image
    Examples:
```

```python
        obj_0 = CLIPRetrieval(objs=objs, query='the yellow and purple
            polka dot pan') # the query is a string
        obj_0 = CLIPRetrieval(objs=objs, query=templates['dragged_obj']) #
            the query is image, stored in templates
    """
    pass

def get_objs_match(objs_list1: list, objs_list2: list) -> tuple:
    """Get correspondences of objects between two lists using the
        Hungarian Algorithm"""
    return (list, list)

## Action Modules
def Pixel2Loc(obj: np.ndarray, masks: np.ndarray) -> np.ndarray:
    """Map masks to specific locations"""
    pass

def PickPlace(pick: np.ndarray, place: np.ndarray, bounds: np.ndarray,
    yaw_angle_degree: float = None, tool: str = "suction") -> str:
    """Pick and place the object based on given locations and bounds"""
    pass

def DistractorActions(mask_obs: list, obj_list: list, tool: str =
    "suction") -> list:
    """Remove observed objects that conflict with the goal object list"""
    pass

def RearrangeActions(pick_masks: list, place_masks: list, pick_ind:
    list, place_ind: list, bounds: np.ndarray, tool: str = "suction") ->
    list:
    """Composite multiple pick and place actions"""
    pass

# Third Level: Connect to Robotic Hardware
def RobotExecution(action) -> dict
    """Execute the robot, then return the execution result as a dict """
    pass

Examples:
------
Use the following examples to understand tools:
## Example 1
# Instruction: Put the checkerboard round into the yellow and purple
    polka dot pan.
def main_1() -> dict:
    """Execute the given instructions of placing the checkerboard round
        into the yellow and purple polka dot pan"""
    image = GetObsImage(obs)
    masks = SAM(image=image)
    objs, masks = ImageCrop(image=image, masks=masks)
    obj_0 = CLIPRetrieval(objs=objs, query='the yellow and purple polka
        dot pan')
    loc_0 = Pixel2Loc(obj=obj_0, masks=masks)
    obj_1 = CLIPRetrieval(objs=objs, query='the checkerboard round',
        pre_obj1=obj_0)
    loc_1 = Pixel2Loc(obj=obj_1, masks=masks)
    action = PickPlace(pick=loc_1, place=loc_0, bounds=BOUNDS)
    info = RobotExecution(action=action)
    return info

## Example 2:
# Instruction: Rotate the {dragged_obj} 150 degrees.
def main_2() -> dict:
    """Execute the given instructions of rotating the {dragged_obj} 150
        degrees"""
```

```python
    image = GetObsImage(obs)
    masks = SAM(image=image)
    objs, masks = ImageCrop(image=image, masks=masks)
    obj_0 = CLIPRetrieval(objs=objs, query=templates.get("dragged_obj"))
    loc_0 = Pixel2Loc(obj=obj_0, masks=masks)
    action = PickPlace(pick=loc_0, place=loc_0, bounds=BOUNDS,
        yaw_angle_degree=150)
    info = RobotExecution(action=action)
    return info

## Example 3
# Instruction: Rearrange to this {scene} then restore.
# Note: for RESTORE operation, direct conduct an inverse operation
def main_3() -> dict:
    """Execute the given instructions of rearranging the objects to match
        the objects in the given scene"""
    image_obs = GetObsImage(obs)
    image_goal = templates.get("scene")
    masks_obs = SAM(image=image_obs)
    objs_obs, masks_obs = ImageCrop(image=image_obs, masks=masks_obs)
    masks_goal = SAM(image=image_goal)
    objs_goal, masks_goal = ImageCrop(image=image_goal, masks=masks_goal)
    row, col = get_objs_match(objs_list1=objs_goal, objs_list2=objs_obs)
    action_1 = DistractorActions(mask_obs=masks_obs, obj_list=col)
    action_2 = RearrangeActions(pick_masks=masks_obs,
        place_masks=masks_goal, pick_ind=col, place_ind=row,
        bounds=BOUNDS)
    action_3 = RearrangeActions(pick_masks=masks_goal,
        place_masks=masks_obs, pick_ind=row, place_ind=col, bounds=BOUNDS)
    actions = []
    actions.extend(action_1).extend(action_2).extend(action_3)
    info = RobotExecution(action=actions)
    return info

## Example 4
# Instruction: Put the yellow and blue stripe object in {scene} into the
    orange object.
def main_4() -> dict:
    """Execute the given instructions of placing the yellow and blue
        stripe object in scene into the orange object"""
    image = GetObsImage(obs)
    masks_obs = SAM(image=image)
    objs_goal, masks_goal = ImageCrop(image=templates['scene'],
        masks=SAM(image=templates['scene']))
    goal = CLIPRetrieval(objs=objs_goal, query='the yellow and blue
        stripe object')
    target = CLIPRetrieval(objs=objs_obs, query=objs_goal[goal])
    loc_0 = Pixel2Loc(obj=target, masks=masks_obs)
    obj_1 = CLIPRetrieval(objs=objs_obs, query='the orange object',
        pre_obj1=target)
    loc_1 = Pixel2Loc(obj=obj_1, masks=masks_obs)
    action = PickPlace(pick=loc_0, place=loc_1, bounds=BOUNDS)
    info = RobotExecution(action=action)
    return info

## Example 5
# Instruction: Put the {dragged_obj} into the {base_obj_1} then
    {base_obj_2}. Finally restore it into its original container.
def mian_5() -> dict:
    masks = SAM(obs_image)
    objs, masks = ImageCrop(obs_image, masks)
    base_obj_1 = CLIPRetrieval(objs, templates['base_obj_1'])
    base_obj_2 = CLIPRetrieval(objs, templates['base_obj_2'],
        pre_obj1=base_obj_1)
```

```
    dragged_obj = CLIPRetrieval(objs, templates['dragged_obj'],
        pre_obj1=base_obj_1, pre_obj2=base_obj_2)
    loc_base_obj_1 = Pixel2Loc(base_obj_1, masks)
    loc_base_obj_2 = Pixel2Loc(base_obj_2, masks)
    loc_dragged_obj = Pixel2Loc(dragged_obj, masks)

    action_1 = PickPlace(pick=loc_dragged_obj, place=loc_base_obj_1,
        bounds=BOUNDS)
    action_2 = PickPlace(pick=loc_base_obj_1, place=loc_base_obj_2,
        bounds=BOUNDS)
    action_3 = PickPlace(pick=loc_base_obj_2, place=loc_dragged_obj,
        bounds=BOUNDS)
    actions = [action_1, action_2, action_3]

    info = RobotExecution(action=actions)
    return info

Begin to execute the task:
------
Please solve the following instruction step-by-step. You should
    implement the main() function and output in the Python-code style.

Instruction: INSERT INSTRUCTION HERE.
```

## A.9 QUANTATIVE ABLATION STUDIES ON PROMPT ELEMENT

We use the same task instruction *Put the polka dot block into the green container.* for all experiments here. Because the outputs of LLMs are somewhat random, we will display the results of three consecutive outputs.

Listing 2: Generation with Only API Definition provided in Prompt. First Trial

```python
def main():
    # Step 1: Get the current observation image
    image = GetObsImage(obs)

    # Step 2: Segment the image to get masks
    masks = SAM(image=image)

    # Step 3: Crop the image based on masks
    objs, masks = ImageCrop(image=image, masks=masks)

    # Step 4: Retrieve the object(s) of interest using CLIP
    query = 'polka dot block' # define the query
    polka_dot_block = CLIPRetrieval(objs=objs, query=query)

    # Step 5: Map masks to specific object locations
    obj_loc = Pixel2Loc(obj=polka_dot_block, masks=masks)

    # Step 6: Get the pick and place locations and action boundaries
    pick_loc = obj_loc # we want to pick up the polka dot block
    place_loc = BOUNDS['green_container'] # the green container is the
        place location
    bounds = BOUNDS['PickPlace_bounds'] # limits of the pick and place
        action

    # Step 7: Perform any necessary distracting actions
    obj_list = ['polka dot block'] # define the goal object list
    masks_obs = np.delete(masks, polka_dot_block, axis=0) # remove the
        polka dot block from the observation masks
    DistractorActions = DistractorActions(mask_obs=masks_obs,
        obj_list=obj_list)
```

```
    # Step 8: Plan and execute the pick and place action
    pick_masks = masks[polka_dot_block]
    pick_ind = [0] # We only want to pick up one object
    place_masks = np.expand_dims(place_loc, axis=0) # expand dims to
        match the pick loc array
    place_ind = [0] # We only have one object to place
    action_seq = RearrangeActions(pick_masks=pick_masks,
        place_masks=place_masks, pick_ind=pick_ind, place_ind=place_ind,
        bounds=bounds)
    action_seq = DistractorActions + action_seq # combine distractor
        actions with pick and place actions
    for action in action_seq:
        result = RobotExecution(action)
        time.sleep(result['time_elapsed']) # wait for the robot to finish
            the action
    return result
```

Listing 3: Generation with Only API Definition provided in Prompt. Second Trial

```
def main():
    # Step 1: Get the current observation image
    obs_image = GetObsImage(obs)

    # Step 2: Use SAM to get segmentation results
    masks = SAM(image=obs_image)

    # Step 3: Crop the image to get object bounding boxes
    objs, masks = ImageCrop(image=obs_image, masks=masks)

    # Step 4: Use CLIP to retrieve the polka dot block
    polka_dot = CLIPRetrieval(objs=objs, query='polka dot block')

    # Step 5: Use CLIP to retrieve the green container
    green_container = CLIPRetrieval(objs=objs, query='green container')

    # Step 6: Map masks to specific locations
    polka_dot_loc = Pixel2Loc(obj=polka_dot, masks=masks)
    green_container_loc = Pixel2Loc(obj=green_container, masks=masks)

    # Step 7: Pick and place the polka dot into the green container
    action = PickPlace(pick=polka_dot_loc, place=green_container_loc,
        bounds=BOUNDS)

    # Step 8: Execute the action using the robotic hardware
    result = RobotExecution(action)

    # Step 9: Save images if execution fails
    if not result['success']:
        SaveFailureImage()

    # Step 10: return the result
    return result
```

Listing 4: Generation with Only API Definition provided in Prompt. Third Trial

```
def main():
    # Step 1: Get the current observation image
    obs_image = GetObsImage(obs)

    # Step 2: Use SAM to get segmentation results
    masks = SAM(image=obs_image)

    # Step 3: Crop the image to get object bounding boxes
```

```
objs, masks = ImageCrop(image=obs_image, masks=masks)

# Step 4: Use CLIP to retrieve the polka dot block
polka_dot = CLIPRetrieval(objs=objs, query='polka dot block')

# Step 5: Use CLIP to retrieve the green container
green_container = CLIPRetrieval(objs=objs, query='green container')

# Step 6: Map masks to specific locations
polka_dot_loc = Pixel2Loc(obj=polka_dot, masks=masks)
green_container_loc = Pixel2Loc(obj=green_container, masks=masks)

# Step 7: Pick and place the polka dot into the green container
action = PickPlace(pick=polka_dot_loc, place=green_container_loc,
    bounds=BOUNDS)

# Step 8: Execute the action using the robotic hardware
result = RobotExecution(action)

# Step 9: Save images if execution fails
if not result['success']:
    SaveFailureImage()

# Step 10: Print the result
print(result)
```

Listing 5: Generation with Only In-Context Examples provided in Prompt. First Trial

```
def main() -> dict:
    """Execute the given instructions of placing the polka dot block into
        the green container"""
    image = GetObsImage(obs)
    masks = SAM(image=image)
    objs, masks = ImageCrop(image=image, masks=masks)
    obj_0 = CLIPRetrieval(objs=objs, query="the polka dot block")
    loc_0 = Pixel2Loc(obj=obj_0, masks=masks)
    obj_1 = CLIPRetrieval(objs=objs, query="the green container")
    loc_1 = Pixel2Loc(obj=obj_1, masks=masks)
    action = PickPlace(pick=loc_0, place=loc_1, bounds=BOUNDS)
    info = RobotExecution(action=action)
    return info
```

Listing 6: Generation with Only In-Context Examples provided in Prompt. Second Trial

```
def main() -> dict:
    # Use object detection to get the object and container locations from
        the RGB image.
    image = GetObsImage(obs)
    masks = SAM(image=image)
    objs, masks = ImageCrop(image=image, masks=masks)
    block_obj = CLIPRetrieval(objs=objs, query='the polka dot block')
    container_obj = CLIPRetrieval(objs=objs, query='the green container')

    # Convert the pixel locations of the objects into robot reference
        frame (in meters).
    block_loc = Pixel2Loc(obj=block_obj, masks=masks)
    container_loc = Pixel2Loc(obj=container_obj, masks=masks)

    # Move the robot's arm to pick and place the block into the container.
    action = PickPlace(pick=block_loc, place=container_loc, bounds=BOUNDS)
    info = RobotExecution(action=action)
    return info
```

Listing 7: Generation with Only In-Context Examples provided in Prompt. Third Trial

```python
def main() -> dict:
    """Execute the given instructions of placing the polka dot block into
        the green container"""
    # Obtain the image of the scene
    image = GetObsImage(obs)

    # Segment the image using Semantic Affinity Module (SAM)
    masks = SAM(image=image)

    # Crop image to obtain individual objects and their masks
    objs, masks = ImageCrop(image=image, masks=masks)

    # Retrieve the polka dot block object using OpenAI's CLIP model
    obj_0 = CLIPRetrieval(objs=objs, query='the polka dot block')

    # Get the location of the green container
    obj_1 = CLIPRetrieval(objs=objs, query='the green container')
    loc_1 = Pixel2Loc(obj=obj_1, masks=masks)

    # Move the polka dot block from its current location to the green
        container
    loc_0 = Pixel2Loc(obj=obj_0, masks=masks)
    action = PickPlace(pick=loc_0, place=loc_1, bounds=BOUNDS)
    info = RobotExecution(action=action)

    # Return the execution information
    return info
```

Listing 8: Generation with both API Definition and In-context Examples provided in Prompt. First Trial

```python
def main() -> dict:
    # first, capture an image of the current scene
    image = GetObsImage(obs)

    # generate object masks using the SAM module
    masks = SAM(image=image)

    # identify the objects in the scene using the masks
    objs, masks = ImageCrop(image=image, masks=masks)

    # use CLIPRetrieval to find the polka dot block and the green
        container
    polka_dot_block = CLIPRetrieval(objs=objs, query="the polka dot
        block")
    green_container = CLIPRetrieval(objs=objs, query="the green
        container")

    # convert the object masks to location coordinates
    polka_dot_block_loc = Pixel2Loc(obj=polka_dot_block, masks=masks)
    green_container_loc = Pixel2Loc(obj=green_container, masks=masks)

    # use the PickPlace function to pick up the polka dot block and place
        it in the green container
    action = PickPlace(pick=polka_dot_block_loc,
        place=green_container_loc, bounds=BOUNDS)

    # execute the action on the robot using the RobotExecution function
        and return the result
    info = RobotExecution(action=action)
    return info
```

