# OpenReview forum: "Instruct2Act: Mapping Multi-modality Instructions to Robotic Arm Actions with Large Language Model"
_ICLR.cc/2024/Conference — Submitted to ICLR 2024_

### Official Review · Reviewer_V9hM · 2023-10-30

**Soundness:** 2 fair
**Presentation:** 2 fair
**Contribution:** 2 fair
**Rating:** 5
**Confidence:** 4

**Summary:**

This study targets the challenge of robotic manipulation tasks utilizing a Large Language Model (LLM) guided by multi-modal instructions. The efficacy of this approach has been tested with various instructions across different benchmarks including VIMABench and the CaP benchmark.

**Strengths:**

1. This system conducts experiments on diverse benchmarks such as VIMABench and the CaP benchmark.

2. Detailed ablation studies are also provided, highlighting the efficacy of 'Library API Full Examples' and different segmentation methodologies.

**Weaknesses:**

1. Important related work RT-2 [1] is missing, likely because it is too recent to be included and discussed.

2. The improvement in performance is not robust. Figure 5 showcases results from evaluations on VIMABench. However, the baseline VIMA-200 performs slightly better than the suggested approach across all tasks. Although the authors argue that VIMA requires large-scale pre-training, their proposed method also depends on large-scale pre-trained foundation models, such as SAM. It would be beneficial for the authors to demonstrate performance gains on tasks where VIMA has not been trained. Furthermore, the performance gain of 3% over PerAct and 1% over CaP in Table 1, is also not significant.

3. Although employing cursor clicks to create point prompts and guide SAM’s segmentation is clever, applying this approach for the Pick and Place task seems unnecessary. If you have the camera extrinsics, you can easily obtain the 3D location of the click by projecting from the image space to the 3D space, given the depth or known height. In this case, there's no need for SAM. In fact, other example tasks should be selected to better highlight the advantages of the proposed method.


[1] Brohan, A., Brown, N., Carbajal, J., Chebotar, Y., Chen, X., Choromanski, K., ... & Zitkovich, B. (2023). Rt-2: Vision-language-action models transfer web knowledge to robotic control. arXiv preprint arXiv:2307.15818.

**Questions:**

This work mentions the utilization of two types of LLMs: text-davinci-003 and LLaMA-Adapter. Could you provide an explanation on the performance differences between these two?

---

> ### Author Response · Authors · 2023-11-22
> **Response to Reviewer V9hM**
>
> >**Q1: Important related work RT-2 [1] is missing, likely because it is too recent to be included and discussed.**
>
> **A1**: We thank the reviewer for this valuable suggestion. RT-2[1] proposes a vision-language-action model that learns from both web and robotics data. In contrast, our method uses LLM to generate logic policy codes, and execute the robotic tasks in a training-free.
>
> >**Q2: However, the baseline VIMA-200 performs slightly better than the suggested approach across all tasks. Although the authors argue that VIMA requires large-scale pre-training, their proposed method also depends on large-scale pre-trained foundation models, such as SAM. It would be beneficial for the authors to demonstrate performance gains on tasks where VIMA has not been trained.**
>
> **A2**:  Thank you for your insightful feedback.
>
> Firstly, regarding the performance comparison with VIMA-200, we agree that the overall performance difference might not be substantial in the tasks evaluated. However, the primary advantage of our method lies in its adaptability, versatility, and training-free. As shown in Table 6 of the main paper, VIMA fails to handle the OOD instructions often, even with 0% SR for Human Intervention. Besides, we have evaluated our method with the (almost) same prompts across different benchmarks, e.g. RLBench, VIMABench and CaP. However, VIMA failed to output reasonable actions when trained in VIMABench while tested in RLBench.
>
> Secondly, we agree that our approach, like VIMA, depends on pre-trained foundation models. However, VIMA requires large-scale pre-training for robotic-specific tasks. Our approach leverages general pre-trained vision foundation models that are not task-specific. This makes our method more scalable and adaptable.
>
> >**Q3: Furthermore, the performance gain of 3% over PerAct and 1% over CaP in Table 1, is also not significant.**
>
> **A3**: Regarding the performance gains over PerAct and CaP, we would like to point out that CaP consumes almost 2.5 times tokens to execute tasks. And they suffer more from the LLM's hallucinations when tasks require strong spatial reasoning. As for PerAct, it needs training on the RLBench, and is unable to handle OOD instructions and to transfer to new tasks, even new benchmarks.
>
> >**Q4: Although employing cursor clicks to create point prompts and guide SAM’s segmentation is clever, applying this approach for the Pick and Place task seems unnecessary. If you have the camera extrinsics, you can easily obtain the 3D location of the click by projecting from the image space to the 3D space, given the depth or known height. In this case, there's no need for SAM. In fact, other example tasks should be selected to better highlight the advantages of the proposed method.**
>
> **A4**: As for the use of cursor clicks in the Pick and Place task, we agree that if camera extrinsics and depth information are known, the 3D location can be computed directly.  However, the strength of our approach is its ability to operate effectively even when such information (the click or extrinsic) is not available or reliable, which is often the case in real-world scenarios.
>
> Additionally, we recognize that objects in the real world are not always uniformly distributed or symmetrical, and the clicked location might be on the edge of the object, which is not suitable for robot grasping. In these cases, if we first use the click to mark the target object, then use the SAM model for semantic segmentation to capture the complete object information, we can determine a more suitable grasping position through simple post-processing.
>
> Lastly, we appreciate your suggestion to select different example tasks to better illustrate the advantages of our method. We plan to conduct additional experiments on tasks where the benefits of our approach are more evident.

---

### Official Review · Reviewer_UX8k · 2023-10-30

**Soundness:** 3 good
**Presentation:** 3 good
**Contribution:** 3 good
**Rating:** 6
**Confidence:** 4

**Summary:**

This paper presents Instruct2Act, a framework that leverages Large Language Models (LLMs) to convert multi-modal instructions to sequential actions for robotic manipulation tasks. It uses pre-defined APIs to access multiple foundation models. Its approach is validated in both simulation and real-world experiments.

**Strengths:**

1. Instuct2Act represents a whole pipeline from utilizing multimodal instructions to actions. This is meaningful in embodied ai domain.

2. It can handle a diverse range of task types.

3. This paper includes suckers and gripper in simulations, and conducts real-world experiments.

**Weaknesses:**

1.	The embodied ai plus LLM develops too fast. It seems that this paper a little bit lacks of novelty. Instruct2Act seems like the combination of VIMA and CaP.

**Questions:**

1.	I think the author should add GPT-4 api and Codellama in ablation?

2.  What’s the main improvement of Instruct2act? Could the authors list some difference between Instruct2act and VIMA, except for using foundation model to detect the objects?

3.	For the franka manipulation, as the paper mentioned: 'Our method is presently limited by the basic action primitives, such as Pick and Place.' So, the franka-based tasks also uses the action primitives?

**Details Of Ethics Concerns:**

No ethic concerns.

---

> ### Author Response · Authors · 2023-11-22
> **Response to Reviewer UX8k**
>
> >**Q1: I think the author should add GPT-4 api and Codellama in ablation.**
>
> **A1**: We greatly thank the reviewer for this valuable suggestion. We add the comparison with GPT-4 API and Codellama as below. Since CodeLLama has a more limited token length, we removed some unrelated function definitions and in-context examples. Same as Table 5, we chose Visual Manipulation (VM) and reported the success rate with 40 runs. Besides, we inspect the error sources due to LLM's incorrect code generation with the case number as described in Appendix A.1. Note that all these LLMs are prompted again without historical information when Python interpreter raises error information. Besides, we test the robustness of these LLMs with the OOD instructions as described in Table, such as appending  ”Cancel the task, stop!” at the end of the original instruction.
>
> | LLM | SR | Syntax error | Attributes hallucination | SR OOD-Instruction |
> | --- | --- | --- | --- | --- |
> | LLama-Adapter | 85.5 | 3 | 2 | 20 |
> | GPT-3.5-Default | 92.5  | 1 | 1 | 90 |
> | CodeLLama-13B | 95 | 0 | 1 | 45 |
> | GPT-4 | 100 | 0 | 0 | 100 |
>
> As shown in the table above, with a stronger LLM, the LLM code generation quality is getting better. As a result, the final performance is getting stronger. And one thing to note is that, since our task could be represented with executable policy python code generation, which directly fails in the strength of CodeLLama, CodeLLama-13B even achieves better performance than GPT-3.5.
> However, when tested with OOD instructions, only GPT-3.5 and GPT-4.0 demonstrate robustness.
>
> >**Q2:  What’s the main improvement of Instruct2act? Could the authors list some difference between Instruct2act and VIMA, except for using foundation model to detect the objects?**
>
> **A2**: We extend our appreciation to the Reviewer for their astute observation. In response, we would like to formulate the differences except for the visual foundation model usage as follows:
>
> - **Training Independence**: Notably, our Instruct2Act operates devoid of any training phase. This stands in stark contrast to VIMA, which necessitates an extensive 650K training trajectories and a robust cluster node infrastructure for its training process.
>
> - **Flexibility in Task Expansion**: In scenarios demanding solutions for novel tasks, Instruct2Act offers inherent scalability. The model can be effortlessly augmented by integrating the requisite function definitions in the python file and supplementing API directives in the prompt. Conversely, VIMA mandates either finetuning or a comprehensive re-training.
>
> - **Out-of-Distribution (OOD) Generalization**: Capitalizing on the capabilities of the LLM, Instruct2Act exhibits commendable adeptness in managing OOD instructional inputs. For instance, when the directive “90 degrees” is substituted with “0.5 radians”, Instruct2Act demonstrates impeccable reasoning prowess to deduce accurate angular values for rotation-oriented tasks, a feat where VIMA-200M conspicuously falls short. Additionally, when the policy is evaluated against alternative benchmarks, such as RLBench, Instruct2Act accomplishes tasks with only minimal alterations. In similar conditions, VIMA-200M, fails to produce meaningful outputs.
>
> >**Q3:  For the franka manipulation, as the paper mentioned: 'Our method is presently limited by the basic action primitives, such as Pick and Place.' So, the franka-based tasks also uses the action primitives?**
>
> **A3**: We thank the reviewer for pointing out this unclearness. The limitation statement of the action primitive is mainly raised by the VIMA simulator. In the evaluation tasks in RLBench, such as opening the drawer, action composing locating, adjusting orientation, and closing the gripper is constructed to grasp the drawer's handler is constructed.

---

### Official Review · Reviewer_KV7t · 2023-11-01

**Soundness:** 3 good
**Presentation:** 4 excellent
**Contribution:** 2 fair
**Rating:** 3
**Confidence:** 4

**Summary:**

The proposed methodology introduces a way to utilize existing off-the-shelf vision and language foundation models for solving robotic tasks. The framework can be conditioned on language and or visual input, and produces Python programs for completing the given task. Experiments in both simulation and the real world demonstrate that the proposed system can outperform baselines.

**Strengths:**

- Well-written. The paper is very well-written and easy to follow. The figures very much aid in understanding the paper.
- Ablations. Ablations are conducted to better understand various design choices of the proposed methodology.

**Weaknesses:**

- Unfair comparison to baselines.
  - CaP
    - The proposed work's claimed distinction with CaP is unclear. The authors write "Unlike existing methods such as CaP (Liang et al., 2022), which directly generates policy codes, our approach generates decision-making actions that can help reduce the error rate when performing complex tasks." But, the proposed methodology also uses LLMs to generate Python programs, so the distinction is unclear. Furthermore, about CaP, the authors write: "However, its capabilities are limited to what the perception APIs can provide, and it struggles with interpreting longer and more complex commands due to the high precision requirements of the code." This seems to be a limitation of the proposed methodology as well, which also relies on a finite number of available APIs. Hence, again, the distinction between the proposed methodology and CaP is unclear.
    - Why is the oracle version of the proposed method used when comparing to CaP, instead of the non-oracle method? This seems to be an unfair comparison.
    - Even then, the performance gap seems trivial. Is there any evidence to suggest that the gap in performance is non-trivial?
  - PerAct
    - The authors write: "For simple 3D tasks, such as stacking objects, we introduced an additional parameter to indicate the current occupancy. For more complex tasks, such as opening a drawer, we added some heuristic movements to ease the execution." PerAct was able to operate without such simplifying assumptions, which makes this comparison unfair.
    - Even then, the performance gap is very small. How do we know this is a non-trivial gap in performance? Is there a trend of tasks / cases where the proposed methodology succeeds and PerAct fails? What is the intuition behind the proposed methodology outperforming PerAct, which was trained directly for the task at hand?
  - Decision Transformer
    - How was comparison to DT done, as DT doesn't take in language instructions?
    - What is the intuition behind the page gap in performance between the proposed methodology and DT?
- Simplistic Tasks.
  - The tasks considered for evaluation seem limited in complexity, e.g. are simply pick-and-place like tasks. These are tasks which can be very easily solved. More complex tasks have not been explored.
  - Furthermore, as the "Generated Policy Code" in Figure 3 depicts, there is an API for pick-and-place, which the system simply calls upon. This simplifies the already simple problem quite a bit, by abstracting away the more difficult low-level control.
  - Finally, a lot of processing / engineering is done to get the system to work (e.g. applying a gray threshold filter followed by a morphological closing operation, a morphological opening operation, Non-Maximum Suppression, etc.). If all of this engineering was required to get a simple task like pick-and-place to work (which is further simplified with the use of a pick-and-place API), then I find it difficult to see how the proposed methodology could be extended to more complex manipulation tasks, thereby limiting its utility.

**Questions:**

- "Although the language models may occasionally generate incomplete or incorrect code, such as missing brackets, punctuation, or mismatched cases, the Python Interpreter can detect these errors and prompt us to generate new code."
  - What is done in these situations? Is the LLM simply prompted a second time, with the hope that the output will not contain incomplete or incorrect code?
- How much prompt engineering went into this when evaluation? Were the prompts the same across tasks, and across methods (ie when comparing to the baselines)?
- How computationally expensive / slow is the framework? Can the tasks be solved in real-time?

---

> ### Author Response · Authors · 2023-11-22
> **Response to Reviewer KV7t [1/2]**
>
> >**Q1: The authors write "Unlike existing methods such as CaP ...**
>
> **A1**: We thank the reviewer for the valuable comment. This comment originates from the observation in Table 6 in the CaP paper.
> The success rate (SR) for task 6 “Pick up the block to the <direction> of the <receptacle-bowl> and place it on the <corner>” is only 72 compared with 100 SR for the naive pick place task. We derive deep into the failure case and find two main failure causes:
>
> 1. Code generation error. The generated codes from CaP could contain some basic errors. One observed failure case is misused (, like “diffs = (points[(:, 1)] - point[1]) ”, which causes SyntaxError.
> 2. LLM Hallucinations. Although the CaP has implicitly taught the LLM about the geometric relationship through the prompt in-context examples, the LLM sometimes still gets confused about geometric reasoning.
>
> To enhance the original CaP [1], we utilized the Instruct2Act methods and added functions as class members in the original CaP classes, and provided function APIs information in the prompts as done in Instruct2Act. We then run 20 trials for task 6 and obtained 18/20, e.g. 90 SR, with 25% improvements.
>
> >**Q2: the authors write: "However, its capabilities are limited to what the perception APIs can provide, and it struggles with interpreting longer and more complex commands due to the high precision requirements of the code." This seems to be a limitation of the proposed methodology as well, which also relies on a finite number of available APIs. Hence, again, the distinction between the proposed methodology and CaP is unclear.**
>
> **A2**: We thank the reviewer for this valuable comment. The representation of "its capabilities are limited to what the perception APIs can provide" comes from the fact that the current version of CaP mainly derives object existence information directly from the simulator's system information with some pre-defined template. In contrast, our method employs visual foundational models to handle complex visual information more flexibly.
>
> >**Q3: Why is the oracle version of the proposed method used when comparing to CaP, instead of the non-oracle method? This seems to be an unfair comparison.**
>
> **A3**: We respectfully disagree with this comment. As stated before, the open-sourced CaP implementation is the version used in the simulation and it assumes that all **object-related information could be accessed by the object's name and id information**. To make a fair comparison, we also implemented an Instruct2Act-Oracle version, where we assume the object detection module returns the ground truth information.
>
> >**Q4: Even then, the performance gap seems trivial. Is there any evidence to suggest that the gap in performance is non-trivial?**
>
> **A4**: Our strengths are from two folders:
>   - Much less token consumption.  As shown in Table 1, our method consumes almost only 40% of tokens when both are evaluated on the CaP benchmarks.
>   -  Fewer LLM Hallucinations with API style. It is evident in the Task Directional Corner, as shown in Table 16, where our method achieves 90 SR while CaP achieves 72 SR.
>
> >**Q5: "PerAct was able to operate without such simplifying assumptions, which makes this comparison unfair." **
>
> **A5**: We thank the reviewer a lot for pointing this out. Firstly, PerAct needs to be trained before testing. They already gain depth reasoning ability during the training. Then, we use a single camera for visual input, while PerAct needs two cameras to induce depth information. Furthermore, the proposed assumption is only a naive walkaround, which could be replaced by the naive projection to 3D space (as suggested by Reviewer V9hM) or directly estimated by other visual models which are invoked also in the API style.
>
> >**Q6:  Is there a trend of tasks / cases where the proposed methodology succeeds and PerAct fails? What is the intuition behind the proposed methodology outperforming PerAct, which was trained directly for the task at hand?**
>
> **A6**: We thank the reviewer for this valuable comment. As shown in Table 15 in our appendix, our method succeeds more with the tasks requiring multiple target objects interaction and multi-step execution, such as push_buttons, we achieved 70 vs PerAct 48. One possible reason is the PerAct is trained with different tasks, some require single-step action, while some require multi-step multi-object interaction, and the learned policy struggles to generalize overall. In contrast, our method abstracts different types of tasks with API-style representation, making it more stable and easy to execute. Moreover, we used the source codes and provided weights in the PerAct webpage, and found that the PerAct is unable to handle the OOD instructions mentioned in the Table 6.

---

> > ### Author Response · Authors · 2023-11-22
> > **Response to Reviewer KV7t [2/2]**
> >
> > >**Q7: How was the comparison to DT done, as DT doesn't take in language instructions?**
> >
> > **A7**: We follow the evaluation pipeline stated in VIMA[1], where DT's initial reward prompt with the VIMA's multimodal task prompt embeddings.
> >
> > [1] Jiang, Yunfan, et al. "Vima: General robot manipulation with multimodal prompts." arXiv (2022).
> >
> > >**Q8: Simplistic tasks with a lot of processing work.**
> >
> > **A8**: This simple action pattern is mainly constrained by the VIMABench simulator, where most tasks are intrinsically the pick-and-place action sequences. However, we also evaluated our method in RLBench, where more complex tasks like opening the drawer and closing jar are used. And we respectfully disagree with the reviewer on the processing/engineering part. Firstly, most of the processing modules mentioned in this paper are very commonly used in CV or the robotic community. Secondly,  we kept the same processing pipeline for all three evaluation benchmarks and found with this simple pipeline, we could achieve promising and robust performance.
> >
> > >**Q9: "Although the language models may occasionally generate incomplete or incorrect code, such as missing brackets, punctuation, or mismatched cases, the Python Interpreter can detect these errors and prompt us to generate new code."What is done in these situations?**
> >
> > **A9**: We thank the reviewer a lot for pointing out this unclearness. In our implementation, we append the last-round generated code lines $C_{t-1}$ with the errors $E_{t-1}$ from the Python Interpreter to construct history information $H$. This $H$ with the task instruction, original in-context examples will be sent to LLM to prompt the LLM to generate the improved code lines with the knowledge of historical errors.
> >
> > >**Q10: Were the prompts the same across tasks, and across methods (ie when comparing to the baselines)?**
> >
> > **A10**:  Yes, we use the same prompts when evaluated across tasks in the same benchmark.
> >
> > >**Q11: How computationally expensive/slow is the framework?**
> >
> > **A11**: When tested on a single NVIDIA RTX 3090Ti, FastSAM takes about 50ms, CLIP operation takes about 40ms, and other general operations take 10ms in total. However, we have to admit that the connection to OpenAI could not be that stable sometimes.

---

### Official Review · Reviewer_F1ER · 2023-11-01

**Soundness:** 3 good
**Presentation:** 2 fair
**Contribution:** 2 fair
**Rating:** 6
**Confidence:** 4

**Summary:**

The paper looks at the problem of generating control programs for robotic control with the help of foundation models. Part of the functionality that can be leveraged in the program are based on visual (SAM) and visual-language (CLIP) foundation models. The programs themselves are obtained by prompting an LLM. The prompt is equipped with external libraries, function descriptions and example programs. The method is tested on benchmarks with text only instructions as well as text-visual instructions and performs competitively. Extensive ablation studies show the importance of individual elements of the prompt and how the choice of foundation models impacts performance. The model shows OOD capabilities, that cannot be achieved by prior methods.

**Strengths:**

- Extensive ablation studies that showcase specific abilities of the model (types of OOD generalization), investigate the role of the foundation models and study the importance of the components of the prompt.

- Removing the dependence of engineered perceptual models from the CaP method by using foundation models.

- Slight improvements over CaP via the different prompting method.

- The methods and results are presented in an understandable manner.

**Weaknesses:**

- Standard Errors: The VIMABench experiments were run over three random seeds for each meta-task but results are reported without any standard errors? In case of the textual prompts benchmarks its also not clear to me why no standard errors on results are reported?

- Choice of baselines: For the VIMABench I find the choice of baselines not insightful. On the one hand the baselines make use of a large set of training trajectories, but on the other hand these methods train policies that choose low-level actions. The presented method chooses action primitives such as "PickAndPlace" but does not need much training data (apart from the examples). I think adapting CaP to the VIMA benchmark would be a more insightful baseline.

- I think it makes sense in the comparison with CaP to compare using an oracle object detector, but one of the main novelties of the work is replacing the perceptual modules of CaP with foundation models. Therefore it would be interesting to see how this affects performance, i.e. just run the method on the CaP benchmark without the oracle.

- Too much detail in related work: I think the related work section is too long and just a list of papers rather than helping to place the work in the research area. I would shorten it and move interesting results from the Appendix to the main paper.

**Questions:**

- Paragraph 4.4: What does RR stand for?

- Are the example programs in the prompt fixed for every task or do they depend on the task ?

---

> ### Author Response · Authors · 2023-11-22
> **Response to Reviewer F1ER**
>
> >**Q1: Standard Errors: The VIMABench experiments were run over three random seeds for each meta-task but results are reported without any standard errors? In case of the textual prompts benchmarks its also not clear to me why no standard errors on results are reported?**
>
> **A1**: We are grateful for the reviewer's insightful suggestion. We reported our method's performance with a success rate following the original VIMA paper, where the standard errors are missing as shown in Table 10 in [1]. To make a direct comparison, we also eliminated that item in our paper. One possible reason for this choice is that the SR is calculated with more than 100 tasks for each meta-task, where the average has already been taken. We are glad to provide the standard errors for these three random seeds.
>
> |    | visual_manipulation | Rotate | scene_understanding | pick_in_order_then_restore | rearrange | rearrange_then_restore |
> |----|---------------------|--------|---------------------|----------------------------|-----------|------------------------|
> | L1 |              0.0039 |     ~0 |              0.0103 |                     0.0308 |    0.0699 |                 0.0155 |
> | L2 |              0.0067 | 0.0194 |              0.0124 |                     0.0476 |    0.0492 |                 0.0168 |
> | L3 |              0.0068 |  0.031 |              0.0144 |                     0.0039 |    0.0413 |            0.033573127 |
>
> >**Q2: Choice of baselines: For the VIMABench I find the choice of baselines not insightful.**
>
> **A2**: To make a fair comparison with other methods on the VIMABench, we mainly follow the VIMA'a paper and use the experiment result from their table.
>
> >**Q3:  I think adapting CaP to the VIMA benchmark would be a more insightful baseline.**
>
> **A3**:  We thank the reviewer for this insightful suggestion. However, one problem with such adaption is that the CaP needs the ground-truth information of the target objects while VIMABench did not directly offer such information. To serve as a walkaround, we extract the GT masks from the simulator and use the center of the mask as the object position. We evaluate the CaP method on the single-step task visual_manipulation,  and multi-step task rearrange_then_restore, and the task requires both memory and saptional reasoning.
> |  | visual_manipulation | rearrange_then_restore | manip_old_neighbours |
> | --- | --- | --- | --- |
> | Ours | 91.3 | 72 | 64 |
> | CaP-Oracle | 100 | 70 | 40 |
>
> As shown in Table, the CaP -Oracle could achieve 100% SR for the simple single-step task. However, it struggles when longer steps, e.g. longer code generation are required. When spatial reasoning, e.g, with directional reasoning and memory mechanism are required, its performance degrades a lot.
>
> >**Q4: Too much detail in the related work.**
>
> **A4**: We thank the reviewer very much for this suggestion. And we will reformulate the paper in the revised version.
>
> >**Q5: What does the RR stand for?**
>
>
> **A5**: We thank the reviewer for pointing out this unclearness. The RR stands for Rotate and Restore task, as described in the caption of Table 4.
>
>
> >**Q6:Therefore it would be interesting to see how this affects performance, i.e. just run the method on the CaP benchmark without the oracle.**
>
> **A6:** We thank the reviewer a lot for this insightful suggestion. We run our method on the CaP benchmark, and the results are shown below. Due to the simple environment setting, our model achieves consistent performance in the task pick_and_place and directional_corner. However, in the tasks where there exist multiple target objects, our method suffers a performance degradation. It is mainly due to the fact that the CLIP calculates the classification scores with the similarity matrix, when there is more than one target object, the classification is affected. At the same time, the task instruction generally did not contain the number information of target objects, thus the generated code sometimes only considers one object.
>
> Besides, we noticed a huge performance gap in the task put_in_line, which is defined as an unseen task in the CaP benchmarks. In our previous implementation, we did not include any information for this task. However, when we adapt some prompts on the line description as done in CaP, we witnessed a boost from 55 to 85.
>
> | Model | pick and place | put in corner | directional corner | stack blocks | put in line |
> | --- | --- | --- | --- | --- | --- |
> | CaP | 88 | 92 | 72 | 82 | 90 |
> | Ours | 100 | 85 | 90 | 85 | 45 |
> | Ours-Oracle | 100 | 95 | 90 | 90 | 55 |
>
> >**Q7: Are the example programs in the prompt fixed for every task or do they depend on the task ?**
>
> **A7:** Yes. They are all the same for all tasks in the same benchmark. There are quite minor modifications for the cross-benchmarks setting, e.g. adding a depth parameter for RLBench.
>
> [1] Jiang, Yunfan, et al. "Vima: General robot manipulation with multimodal prompts." arXiv (2022).

---

> ### Comment · Reviewer_F1ER · 2023-11-23
>
> Thank your for the detailed response and running the additional experiments.
>
> A1. Thank you, it seems that standard errors are not reported since they are too small.
>
> A3. I think this results strengthens the evidence that the approach improves over CaP.
>
> A6. I think this additional experiment adds value to the paper by showing that performance can largely be maintained even without the Oracle.
>
> All of my main concerns have been addressed. Accordingly I will adjust my score.

---

### Meta-Review · Area_Chair_cC5g · 2023-12-10

**Metareview:**

Synopsis: This paper presents a system to leverage LLMs and visual foundation models to perform tabletop manipulation tasks. The task prompt is used to generate python code using the LLM, which then interactively queries visual foundation models to perform the task.

Strengths:
+ Lots of empirical evaluations showing the performance on a variety of tasks
+ The use of visual foundation models eliminates the need for engineered solutions

Weaknesses:
- The paper does not convincingly articulate what is technically novel w.r.t. previous work, such as code-as-policies. The stated innovations seem minor, and the empirical results do not convince the reader that the stated innovations provide a significant boost.
- While the results include a few real world tasks, this is significantly fewer than the state of the art, again, including code-as-policies.
- The choice of using an oracular detector is questionable.

**Justification For Why Not Higher Score:**

Having read the paper and the reviews in reasonable depth, I am not convinced by the stated innovations over the state of the art. The empirical evaluations (e.g., comparisons to CaP, use of oracular detectors, etc.) raise more questions, as in the author response stage. While the authors tried to perform a few additional experiments to allay these concerns, I do not think they were convincing.

**Justification For Why Not Lower Score:**

N/A

---

### Decision · Program_Chairs · 2024-01-16

Reject